# Recyclable Self-Healing Polyurethane Cross-Linked by Alkyl Diselenide with Enhanced Mechanical Properties

**DOI:** 10.3390/polym11050773

**Published:** 2019-05-01

**Authors:** Yuqing Qian, Xiaowei An, Xiaofei Huang, Xiangqiang Pan, Jian Zhu, Xiulin Zhu

**Affiliations:** 1State and Local Joint Engineering Laboratory for Novel Functional Polymeric Materials, Jiangsu Key Laboratory of Advanced Functional Polymer Design and Application, Department of Polymer Science and Engineering, College of Chemistry, Chemical Engineering and Materials Science, Soochow University, Suzhou 215123, China; 20164209031@stu.suda.edu.cn (Y.Q.); 20164009027@stu.suda.edu.cn (X.A.); huangxiaofei314@163.com (X.H.); xlzhu@suda.edu.cn (X.Z.); 2Jiangsu Litian Technology Co. Ltd., Rudong County, Jiangsu 226407, China; 3Global Institute of Software Technology, No 5. Qingshan Road, Suzhou National Hi-Tech District, Suzhou 215163, China

**Keywords:** dynamic structures, diselenide, self-healing

## Abstract

Dynamic structures containing polymers can behave as thermosets at room temperature while maintaining good mechanical properties, showing good reprocessability, repairability, and recyclability. In this work, alkyl diselenide is effectively used as a dynamic cross-linker for the design of self-healing poly(urea–urethane) elastomers, which show quantitative healing efficiency at room temperature, without the need for any catalysts or external interventions. Due to the combined action of the urea bond and amide bond, the material has better mechanical properties. We also compared the self-healing effect of alkyl diselenide-based polyurethanes and alkyl disulfide-based polyurethanes. The alkyl diselenide has been incorporated into polyurethane networks using a para-substituted amine diphenyl alkyl diselenide. The resulting materials not only exhibit faster self-healing properties than the corresponding disulfide-based materials, but also show the ability to be processed at temperatures as low as 60 °C.

## 1. Introduction

Thermosets are widely used in aerospace [1,2], the electronics industry [3], machinery and other fields [4,5,6] due to their network or body structure which results in good physical, mechanical and application properties [7,8]. Thermosets are also the materials of choice for numerous applications because of their dimensional stability, and creep/chemical resistance [9]. To meet growing demands in society, thermosets such as epoxy resin (EP) and polyurethane (PU) are widely employed. However, because of the stability of the thermoset’s cross-linked structure, it cannot be recycled after it is formed, thus causing problems such as resource waste and environmental pollution [10,11].

By the introduction of exchangeable chemical bonds, plasticity in cross-linked polymer networks is introduced, leading to dynamic cross-links. Recently, the introduction of dynamic covalent bonds into polymer networks has been used as an attractive approach towards the design of various intrinsically self-healing polymer systems [12,13,14]. The Diels–Alder reaction [15], transesterification [7], olen metathesis [16,17], radical reshuffling [18], imine [19] or hydrazine formation [20], siloxane equilibration [21], thiol–nanoparticle exchange [22] and aliphatic disulfide exchange [23] are some examples of reversible covalent chemistries used for the design of self-healing polymers. The idea behind this is to reconnect the chemical cross-links which are broken when a material fractures, restoring the integrity of the material. This is expected to provide polymers with enhanced lifetime and resistance to fatigue. However, the stronger nature of dynamic covalent bonds compared to noncovalent ones offers the possibility to obtain self-healing polymer networks with superior mechanical strength.

Selenium and sulfur tend to form dimers, as in the case of peroxides or disulfides. However, in comparison to disulfide bonds, diselenide bonds possess a lower bond energy (diselenide bonds: 172 kJ mol^−1^; disulfide bonds: 240 kJ mol^−1^) [24]. This suggests that diselenide bonds can be more dynamic, and that the exchange reaction might happen under much milder conditions, which could help to overcome the disadvantages of disulfide exchange reactions. A lot of research has been done on the use of disulfide to prepare self-healing materials and it has achieved good healing effects. This increased interest in diselenide amongst researchers [25,26,27,28,29], resulting in the development of self-healing materials. Xu et al. proposed the introduction of aliphatic diselenide compounds to produce healable materials in which the exchange of diselenide moieties is triggered by the application of UV and visible light [30,31]. Zhu et al. demonstrated the ability of aromatic diselenide to exchange faster than aromatic disulfides on account of the barriers that are around 25–30 kJ/mol lower in the diselenide reaction [32], than in the disulfide reaction. However, while these systems can have better healing efficiency and mechanical properties, there is still room for improvement, such as reduced healing time and improved mechanical properties.

Therefore, the present work reports the ability of diselenium alkyl systems to incorporate dynamic properties to cross-linked materials, and the ease of preparation from commercially available starting materials makes this system broadly applicable in a wide number of industrial sectors where poly(urea–urethane)s are already used. To achieve this, the alkyl diselenide was synthesized and characterized. Alkyl diselenide has been compared to alkyl disulfide compounds, showing a better self-healing effect and healing efficiency can reach 100%. in a shorter time Finally, alkyl diselenide containing soft polyurethane thermosets are synthesized and their potential reprocessability and self-healing abilities are investigated.

## 2. Experimental

### 2.1. Material and Reagents

Isophorone diisocyanate (IPDI, 1), dibutyltin dilaurate (DBTDL) and 4-aminobenzylamine were purchased from Acros (Shanghai, China). Poly(propylene glycol)s (PPG), and 2 (*M*_n_ = 2000), 3 (*M*_n_ = 3000) and 4-chlorobutyryl chloride were purchased from Adamas (Shanghai, China). Dry tetrahydrofuran (THF) was collected fresh from an Innovative Technology PS-MD-5 solvent purification system. Tetrabutylammonium hydrogen sulfate (Bu_4_NHSO_4_) was provided from TCI (Shanghai, China). γ-Butyroselenolactone was prepared according to reference [33]. Other chemicals were purchased from Shanghai Regant Co. Ltd. China and used without treatment.

### 2.2. Synthesis of γ-Butyrothiolactone

NaSH (14.2 g, 200 mmol) and Bu_4_NHSO_4_ (1.7 g) were slowly added to a solution of appropriate 4-chlorobutyryl chloride (28.2 g, 200 mmol) in toluene (250 mL). The mixture was vigorously stirred at 25 °C for 6–12 h under argon. The reaction mixture was washed with aq 5% NaHCO_3_ (100 mL) and dried by anhydrous Na_2_SO_4_ overnight. Toluene was removed in vacuo. The resulting residue was distilled under reduced pressure to give γ-butyrothiolactone. Yield: 12.65 g, 62%. The structure was confirmed by ^1^H NMR spectroscopy (^1^H NMR, 300 MHz, CDCl_3_), δ 3.42 (t, 2H), 2.50 (t,2H), 2.18 (m,2H)).

### 2.3. Synthesis of Diselenide- and Disulfide-Functionalized Diamine Cross-Linker

The diselenide-functionalized diamine cross-linker was synthesized by a ring opening reaction of γ-butyroselenolactone with 4-aminobenzylamine. γ-Butyroselenolactone (1.23 g, 0.82 mmol) and 4-aminobenzylamine (1.00 g, 0.82 mmol) were dissolved in 30 mL dry THF and headed at 40 °C for 2 h. The THF was removed in vacuo and the yellow solid product was washed and dried. The yield is close to 100%. The structure was confirmed by NMR and HRMS (C_24_H_26_O_2_N_2_Se_2_): m/z 565.0613 (M + Na^+^, calcd 565.0591), (Appendix A). The disulfide-functionalized diamine cross-linker was synthesized according to the same procedure by using γ-butyrothiolactone as the reactant. The structure was confirmed by NMR and HRMS (C_24_H_26_O_2_N_2_S_2_): m/z 469.1693 (M + Na^+^, calcd 469.1702), (Appendix A).

### 2.4. Synthesis of Di-Isocyanate-Terminated Urethanes 4

Di-isocyanate-terminated PPG was synthesized according to the procedure shown in Scheme 1. A mixture of PPG (2000 g/mol) (30 g, 15 mmol) and IPDI (6.66 g, 30 mmol) was fed into a 500-mL glass reactor equipped with a mechanical stirrer and vacuum inlet. The mixture was degassed by stirring under vacuum while heating at 80 °C for 60 min. Then dibutyltin dilaurate (DBTDL) (50 ppm) was added and the mixture was further stirred under vacuum at 70 °C for 60 min. The reaction was monitored by FTIR spectroscopy (Appendix A), where the appearance of new bands corresponded to the carbonyl group of the urethane moiety at 1708 cm^−1^ and amide II can be observed at 1519 cm^−1^. Moreover, a decrease and displacement of the NCO stretching band from 2257 to 2269 cm^−1^ can be observed, which was used as criteria to establish that the reaction was finished. The resulting tris-isocyanate-terminated prepolymer was obtained in the form of a colorless liquid and stored in a tightly closed glass bottle.

### 2.5. Synthesis of Tri-Isocyanate-Terminated Urethanes 5

Tri-isocyanate-terminated PPG was synthesized according to the procedure shown in Appendix A. PPG (3000 g/mol) (30 g, 10 mmol) was fed into a 500-mL glass reactor equipped with a mechanical stirrer and vacuum inlet. The mixture was degassed by stirring under vacuum while heating at 80 °C for 60 min. Then DBTDL (50 ppm) and IPDI (6.66 g, 30 mmol) were added and the mixture was further stirred under vacuum at 70 °C for 50 minutes. The reaction was monitored by FTIR spectroscopy (Appendix A). The appearance of new bands corresponded to the carbonyl group of the urethane moiety at 1722 cm^−1^ and amide II can be observed at 1531 cm^−1^ after the reaction. Moreover, a decrease and displacement of the NCO stretching band from 2247 to 2266 cm^−1^ can be observed, which was used as criteria to establish that the reaction was finished. The resulting tris-isocyanate-terminated prepolymer was obtained in the form of a colorless liquid and stored in a tightly closed glass bottle.

### 2.6. Synthesis of Alkyl Diselenide-Based Polyurethanes

The isocyanate-terminated urethanes 4 and 5 were mixed in a 100 mL open mold. Then, a solution of diselenide diamine or disulfide diamine in THF was degassed in vacuum for 1.5 h. The overall NCO/NH_2_ ratio of the mixture kept at 1.0. The curing was carried out for 24 h at 60 °C and the process could be easily monitored by FTIR spectroscopy, the results of which are shown in Appendix A for diselenide and disulfide containing polyurethane, respectively. 

### 2.7. Characterization

Nuclear magnetic resonance (NMR) spectra were recorded on a Bruker 300 MHz nuclear magnetic resonance instrument using CDCl_3_ or DMSO as the solvent and tetramethylsilane (TMS) as an internal standard. Fourier transform infrared (FTIR) spectra were measured on a Bruker TENSOR 27 FTIR spectrometer (KBr disk). UV-vis spectra were recorded on a Shimadzu UV-3150 spectrophotometer (Shimadzu China, Shanghai, China). Differential scanning calorimetry (DSC) was performed with heating and cooling at a rate of 10 °C/min with the temperature ranging from −100–50 °C on a Q200 differential scanning calorimeter (TA Instruments). The glass transition temperature (*T*g) was recorded on the second cycle of a heat/cool experiment. Thermogravimetric analysis (TGA) was performed on PerkinElmer Pyris 1 instruments. Samples were run in platinum TGA pans at a ramp rate of 10 °C/min from 30 to 800 °C under the nitrogen atmosphere. Mass Spectrometry (MS) was measured on a Bruker micro TOF-QIII spectrometer (Karlsruhe, Germany) using acetonitrile and dimethyl sulfoxide as the solvent. All the tensile tests were carried out in air, at room temperature, using a universal tensile machine (KJ-1065B, Kejian-tech) with a 50-N loading cell. The cross-head speed of the tensile measurements was 100 mm/min, unless specified otherwise. In order to avoid systematic failure in the vicinity of the clamps, all the samples were cut using a dumbbell-shaped die cutter.

## 3. Results and Discussion

### 3.1. Synthesis of Polyurethane

Polyurethane with a diselenide dynamic bond was synthesized via a simple procedure according to Scheme 1. The diselenide-functionalized diamine was synthesized through ring-opening and the oxidative coupling reaction of γ-butyroselenolactone with 4-aminomethylaniline. Based on our previous work, methylamine groups in 4-aminomethylaniline could be used to react with γ-butyroselenolactone, while another arylamine was inert in such a reaction [34]. Such a highly selective reaction between alkyl amine in 4-aminomethylaniline with γ-butyroselenolactone means such a reaction could be carried out without any protection–deprotection procedures (Scheme 1a). The diselenide-functionalized diamine product could be obtained with a high yield (close to 100%) under mild conditions. The structure of the product was confirmed by NMR and HRMS. For comparison, the analog compound with disulfide-functionalized diamine was synthesized using a similar procedure. NMR and HRMS confirmed the structure of the product.

Polyurethanes were synthesized starting from commercially available poly(propyleneglycol) (PPG) resins 2 and 3, having an average molecular weight of 2000 and 3000 g/mol, respectively. They were treated with isophorone diisocyante (IPDI) using dibutyltin dilaurate (DBTDL) as a catalyst to modify the terminal groups into isocyanate. Thus, bi- and tri-isocyanate-terminated (4 and 5 in Scheme 1) PPGs were obtained respectively (Scheme 1b). The reaction was monitored by FTIR spectroscopy. Typical spectra are shown in Appendix A. Then, the mixture of di-isocyanate-terminated PPG and tri-isocyanate-terminated PPG with molar ratios of 0:1, 0.25:1, 0.5:1, 0.75:1, and 1:1 was reacted with diselenide-functionalized diamine with the molar ratio of NCO/NH_2_ = 1:1 to achieve the respective polyurethanes (6a in Scheme 1b). The mixture was cured in a mold at 60 °C for 24 hours. The curing process could be easily monitored by FTIR spectroscopy. A typical spectrum is shown in Appendix A, where the isocyanate stretching band at 2264 cm^−1^ completely disappeared and a new band in the form of a shoulder corresponding to the urea appeared at 1650 cm^−1^ after the reaction. Similarly, the FTIR spectra of disulfide-based systems (7a in Scheme 1) presented the same spectroscopic features (Appendix A). The obtained polyurethanes were of a light orange color with hard elasticity (Scheme 1c), but the color of the disulfide-functionalized diamine cross-linked polyurethane is deeper than that of the diselenide-functionalized diamine cross-linked polyurethane.

### 3.2. Mechanical Properties and Healing Properties

In order to optimize the mechanical properties of materials, polyurethanes with different reactants molar ratio were synthesized by adjusting the ratio of di-isocyanate-terminated PPG to tri-isocyanate-terminated PPG as follows: 0:1, 0.25:1, 0.5:1, 0.75:1, and 1:1. The obtained materials were cut in the form of dumbbell-shaped specimens for tensile strength measurements. The results are shown in Figure 1. 

All the materials showed relatively high stress, e.g., higher than 5 MPa at break, with strains larger than 550%, which was much higher than the materials reported by our previous paper [32] and Xu’s paper [30]. Also, the toughness of the materials was calculated and the results are shown in Table 1. The polyurethane synthesized from bis-isocyanate-terminated urethanes and tris-isocyanate-terminated urethanes at a molar ratio of 0.25:1 shows the highest toughness, 41.63 MPa. The introduction of the amide bonds into the current material would be helpful to improve such properties due to the easy formation of hydrogen bonding between such groups [35,36]. Furthermore, it could be found that the mechanical strength of the material decreases with increasing the proportion of di-isocyanate-terminated PPG. Such a result may be due to the difference in the cross-linking density inside these materials. Although there is no direct data of cross-linking density in the current results, increasing the ratio of di-isocyanate-terminated PPG in the mixture would evidently decrease the cross-linking density of the material, leading to a decrease in its mechanical properties.

The healing efficiency of these five materials synthesized from different proportions of di- and tri- isocyanate-terminated PPG were also examined. The healing efficiency is calculated from the area ratio of the pristine to the healed sample, where the area refers to the area enclosed by stress and strain. Dumbbell-shaped specimens were cut off in the middle part, put in close contact and healed for 24 h at 25 °C. Tensile test results (Appendix A) show that with the increase of the proportion of di-isocyanate-terminated PPG, the healing efficiency of the final material is higher (Figure 2). When the material contains only tris-isocyanate-terminated urethanes, the healing efficiency is the lowest, and the healed samples recovered 9% of their initial mechanical properties. However, when the same proportion of bis-isocyanate-terminated urethanes and tris-isocyanate-terminated urethanes is used, the healing effect of the material can reach 100%.

Thus, the performance of alkyl diselenide-based polyurethanes synthesized from bis-isocyanate-terminated urethanes and tris-isocyanate-terminated urethanes at a molar ratio of 1:1 were further investigated. Firstly, the self-healing time was investigated by cutting specimens in half and then mending them by simple contact at room temperature for different periods of time. The results were summarized in Figure 3A. After 10 and 30 min, 28 and 75% of the initial mechanical property was recovered, respectively. While 100% of its initial mechanical property was recovered after 1 h, the healing effect of the dumbbell-shaped specimen can be obviously seen from the picture (Appendix A). The healing rate of the material is significantly higher than that of a recently published paper [37]. These results show that the self-healing efficiency of the obtained material is very satisfactory in common applications. The self-healing efficiency of diselenide-derived polyurethanes was faster than their disulfide counterparts (Figure 3B). Around 80% of its initial mechanical properties could be recovered after 3 hours for disulfide-based polyurethane, which was similar to the materials reported [28,29]. This result is consistent with the paper previously reported by An et al. In comparison to disulfide bonds, diselenide bonds possess a lower bond energy (disulfide bonds: 240 kJ mol^−1^; diselenide bonds: 172 kJ mol^−1^), which suggests that diselenide bonds can be more dynamic and the exchange reaction can happen faster.

The effect of temperature on healing efficiency was also explored. The specimens were cut in half and then mended by simple contact at different temperatures for 10 min. The tensile test results showed that temperature was important for the self-healing of the current material (Figure 3C). Only 28% of the initial mechanical property was recovered at 25 °C after 10 min, while the healing efficiency increased to 32, 56 and 100% at 30, 40, 50 °C, respectively. Increasing the temperature to 60 and 70 °C, the healing efficiency remained at 100% due to more time. Thus, the healing efficiency increased with increasing of temperature due to the acceleration of the exchange rate of dynamic covalent bond with temperature. 

The sample was subjected to a cyclic tensile test. A slight decrease in both modulus and stress was detected in the tensile measurement immediately after the cyclic tensile test. However, the sample recovers all of its initial mechanical properties after a waiting period of 5 min, as evidenced by an overlap of the tensile curves (Figure 3D). These results indicated that chain sliding does not occur under the large deformation. The length of the original sample and the sample that waited for five minutes after the cyclic tensile test were compared. The latter has about 20% deformation compared to the former. This proved the material has good performance in its own resilience.

### 3.3. Reprocessing of Diselenide Polyurethanes

Due to the existence of dynamic covalent bonds, the ability of the diselenide to be reprocessed at moderate temperatures was studied. Firstly, the samples were ground down to use as raw substances for compression molding (Figure 4A). They can then be reprocessed at 60 °C under pressure (10 bar) to form a complete film. After reprocessing the polymer materials, the morphology and mechanical properties of the diselenide bond containing the materials could regenerate at approximately 100% (Figure 4B). The self-healing properties of the processed material were also examined. Similarly, the processed materials were cut in the form of dumbbell-shaped specimens for tensile strength measurements. The healed specimens were tested after being cut in two parts, put in close contact and healed for 12 h and 24 h at room temperature. The tensile test results show that the recovery of the material was already 50% at 12 h. While the recovery of the material was 100% at 24 h. (Figure 4B). The Se–Se bond is sensitive and, during the reprocessing process, the surface of the material may be oxidized, resulting in a decrease in self-healing properties. These results show that the diselenide thermoset materials present reprocessable, recyclable and self-healing features. Such polyurethanes with strong mechanical and self-healing properties are recyclable and can also be synthesized in large amounts (Figure 4C). It has strong potential to be commercially produced in the future.

Thermogravimetric analysis (TGA) showed alkyl diselenide-based polyurethanes with stability up to 300 °C (Figure 5A), which could be attributed to the presence of diselenide species. From the differential scanning calorimeter (DSC), the *T*g value is below zero. (Figure 5B).

Alkyl diselenide-based polyurethanes in large deformation can still return to their original state. Therefore, in order to study the deformation recovery performance of the materials, we conducted a cyclic tensile test. The maximum strain was 150, 200, 300, and 400% cyclic tensile test, respectively. When the maximum strain is 150% (Figure 5C), the five curves basically coincide. When the maximum strain is 400% (Figure 5D), the first cyclic curve is slightly different from the following four curves, but the test curves of the following four times are basically coincident. This proves that the material has better recoverability.

## 4. Conclusions

In summary, a new poly(urea–urethane) thermoset elastomer with alkyl diselenide cross-links has been designed. The materials containing alkyl diselenide moieties show excellent healing abilities without the need for any external intervention such as heat or light. At the same time, the material also has good mechanical properties. We have demonstrated that such networks can be reshaped to the desired form by just placing them in a hot press, without losing their initial mechanical properties. The simplicity of the synthesis process, as well as the wide variety of dynamic chemistries available, makes the method useful for the development of advanced polymeric materials.

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
