# Peer review of "Recyclable Self-Healing Polyurethane Cross-Linked by Alkyl Diselenide with Enhanced Mechanical Properties"

_polymers, 2019, doi:10.3390/polym11050773_

Reviewer 1 Report

*The work is a variant of polyurethane based self-healing systems that contain dynamic bonding. The self-healing mechanism relies on the S-S and Se-Se linkers. And the polymer backbone allows for enough chain mobility for self-healing to occur within the polyurethane segments. Although there are relevant papers about Se-Se and S-S bearing polyurethanes, the paper is still a detailed study and can be published. 

*The polymers were also chopped and remolded. Thus, this process is recycling thus the title of the paper should also reflect this process.  

*Images of cut-healed specimens can be added as a supporting info figure.

*The amount of applied pressure or the method that used for recycling process should be described.

*There are also other polyurethane based systems that have Se-Se and S-S dynamic bonds and could be cited.

Author Response

Thanks for the valuable comments. The detailed responses for these comments has been summarized in attached document.

Reviewer 2 Report

This work describes the self-healing of PUs derived by alkyl diselenide comparison with disulfide moiety. However, the work needs to more supplements.

So I recommend that publish after minor revision to this article that needs more complementations. The specific points of insufficiency are outlined below:

1. The authors citated the several references(~about 36 reference). However, most of references(~34references) are concentrated on the introduction part. Then, most of results are new findings?(only 2 reference)

In my opinion, reference paper studies are not sufficient.

In page 2 line 90, “1H NMR” should be “1H NMR”.

In page 3 line 129, there is usually defined that the glass transition temperature is defined at 2nd run not the 3rd run [as the author claimed heat (1st heat)-cool (1st cool)-heat (2nd heat))

In page 4, the morphology characterization and enough explanation of morphological difference within diselenide and disulfide in Scheme 1 (C) through discussion.

In page 4 line 146-149, “γ--“ is correct? Not “γ-“ ?

In page 5 line 153-156, there should be the method of HRMS in the experimental section and HRMS spectra in supporting information. Also, there are no other molecular weight of Polyurethane any where and only monomer’s. Because molecular weight of polymer is important for the mechanical properties as well.

Through Fig 1~Fig 4, I recommended to calculate the toughness (~MJ/m3). Because not only the Stress and Strain but toughness is the characteristic factor of mechanical properties. And recommended the fig 2, should be  supported by the Stress-Strain(S-S) curves also.

 In page 6 line 199-200, the dumbbell type specimens were used, I recommend supplementation that the self-healing sample before/after sample picture in the supporting information.

 In page 8 line 268, the author claimed that “dynamic mechanical analysis(DCS)”, where could I find related data? I assume that is DSC (differential scanning calorimeter) is correct. If the claimed that is Tg, they should be indicate in the Fig 5 B. Furthermore, in “Fig 5B, Y-axis”, it has to be defined exo up or endo down for the heat flow direction, clearly.

In page 8 Figure 4 B, the mistypo of legend for sample should be revise. (Reprocessed semple -> sample)

In supporting sections, Figure S3~S6, I recommend the assign of peak corresponding to each functional groups within polymers, respectively.

Author Response

Thanks for your valuable comments. The detailed response for all of the comments has been summarized in attached document. Thanks.
